# Overview of the Effect of Different Regenerative Materials in Class II Furcation Defects in Periodontal Patients

**DOI:** 10.3390/ma15093194

**Published:** 2022-04-28

**Authors:** Gerasimos Odysseas Georgiou, Francesco Tarallo, Enrico Marchetti, Sergio Bizzarro

**Affiliations:** 1Department of Periodontology, Academic Centre for Dentistry Amsterdam (ACTA), University of Amsterdam and Vrije Universiteit, 1081 LA Amsterdam, The Netherlands; gerodgeorgiou@hotmail.com (G.O.G.); s.bizzarro@acta.nl (S.B.); 2Department of Life, Health and Environmental Sciences, University of L’Aquila, Coppito, 67100 L’Aquila, Italy; francesco.tarallo93@gmail.com

**Keywords:** regeneration, furcation, periodontitis, periodontal defects

## Abstract

The aim of this review was to give an overview of the outcomes of the use of different regenerative materials to treat molars with class II furcation defects in patients with periodontitis in comparison with open flap debridement (OFD). A search of five databases (PubMed-Medline, Embase, Cochrane, Scopus and Web of Science) was conducted up to and including January 2022. According to the PICOS guidelines, only randomized control trials (S) considering periodontal patients with at least one molar with a class II furcation involvement (P) treated with regenerative materials (I) in comparison to OFD as control treatment (C) and a minimum follow-up period of 6 months were included. Vertical clinical attachment level (VCAL) was considered as the primary outcome (O), while horizontal clinical attachment level (HCAL), horizontal probing depth (HPD) and vertical probing depth (VPD) were considered as secondary outcomes. The search through the databases initially identified 1315 articles. Only 25 of them met the eligibility criteria and were included. The studies were grouped in four macro-categories according to the material used: absorbable and non-absorbable membranes, blood derivatives and a combination of different materials. The greater part of the included studies reported a statistically significant difference in using regenerative materials when compared to OFD. The blood derivatives groups reported a range of mean changes in VCAL of 0.86–4.6 mm, absorbable membrane groups reported −0.6–3.75 mm, non-absorbable membranes groups reported −2.47–4.1 mm, multiple materials groups reported −1.5–4.87 mm and enamel matrix derivatives reported a mean change in VCAL of 1.45 mm. OFD showed a range of mean VCAL changes of −1.86–2.81 mm. Although no statistical analysis was performed, the use of regenerative materials may be considered moderately beneficial in the treatment of molars with grade II furcation involvement. However, the substantial heterogeneity in the protocols’ design does not allow us to draw definitive conclusions. In addition, low levels of evidence for morbidity and patient-centered outcomes were reported.

## 1. Introduction

Periodontitis is a chronic, multi-factorial inflammatory disease, caused by an inflammatory reaction by the host to a dysbiotic subgingival microflora, which can be modified by genetic and lifestyle factors, and it results in the loss of tooth-supporting apparatus, the connective tissue attachment and alveolar bone [1,2,3,4]. Worldwide, periodontitis affects about 20–50% of the global population [5].

When left untreated or not successfully treated, periodontitis can eventually cause the loss of the tooth, gradually leading to a loss of both chewing and aesthetical function. The progression of the bone loss seems to be more prevalent in multi-rooted teeth, particularly the upper and lower molars [6]. The treatment of the furcation regions of the molars is often less effective when compared to the non-molar teeth, because of their complex anatomy and the possible presence of other abnormalities as enamel projections, enamel pearls or grooves [7,8,9,10,11]. A retrospective cohort study reported that 58.1% of the patients lost at least one molar over 10 years of supportive periodontal treatment after active periodontal therapy [12].

To improve both treatment outcomes and the prognosis of the molar teeth, different resective surgical approaches have been proposed—more specifically, root resection, hemi-section or tunnelization. However, these treatments showed a considerable amount of failures (25–58%) and a high level of morbidity and costs [13].

Alternatively, the regeneration of the lost periodontal tissues is also another treatment option. According to the Glossary of Periodontal Terms, regeneration is the “reproduction or reconstitution of a lost or injured part in a manner similar or identical to its original form. In periodontics, it refers to the formation of new bone, cementum, and a functionally-oriented periodontal ligament at a site deprived of its original attachment apparatus” [14]. To achieve this goal, different biomaterials have been investigated.

Guided Tissue Regeneration

In guided tissue regeneration (GTR), the gingival epithelium is excluded from the healing wound in order to allow a selective repopulation of the root surface by cells of the periodontal ligament and alveolar bone. This will prevent the rapid proliferation of the sulcular epithelium and the forming of a long junction epithelium on the root surface [15]. To achieve this, different sorts of barriers (membranes) have been used [15,16,17].

The main characteristics for a GTR membrane should be:Biocompatibility to allow integration with the host tissues without eliciting inflammatory responses.Proper degradation profile to match with the new tissue formation.Adequate mechanical and physical properties to allow its placement in vivo.Sufficient sustained strength to avoid membrane collapse and perform their barrier function [18].

A major distinction among membranes is as follows:Non-absorbable barriers

The mostly used non-absorbable membranes are made of tetrafluoroethylene or expanded polytetrafluoroethylene. Other materials were rubber dam, resin–ionomer barrier, a barrier made out of knitted nylon fabric mechanically bonded onto a semipermeable silicon membrane and coated with collagen peptides, and Millipore filter [19]. The main issues related to non-absorbable barriers are the high chance of bacterial contamination and the need for a second surgery for removal.

Absorbable barriers

Absorbable membranes offer better biocompatibility than the non-absorbable ones but low to no control over the regenerative healing. In fact, the degradation process starts immediately after the placement in the tissues and the rate of absorption may vary among patients. Absorbable membranes are made of collagen and various derivates of collagen such as dura mater, cargile membrane, oxidized cellulose, laminar bone, connective tissue graft, polyglycolic acid and polylactic acid [20,21].

Grafts

Graft materials, or bone substitutes, are used mainly to stabilize the blood clot in the alveolar bone defect and allow regeneration of the periodontal tissues. Graft materials should present one of the following characteristics to promote regeneration:They contain bone-forming cells (osteogenesis).They function as a scaffold for bone formation (osteoconduction).They contain biological substances in their matrix that induce bone formation (osteoinduction).

Graft materials can be subdivided in four categories:Autogenous: Grafts obtained by the patient, harvested both from intraoral and extraoral sites, consisting of cortical bone or cancellous bone and marrow.Allogeneic: Grafts of human origin. Three types of bone allografts are used in periodontics, namely, demineralized freeze-dried bone, non-demineralized freeze-dried bone and frozen iliac cancellous bone.Xenogeneic: Graft from a non-human donor, mainly from bovine or porcine origin.Alloplastic materials: Synthetic or inorganic implant materials which are used as substitutes for bone grafts [21,22].

Enamel Matrix Derivative

This is a purified fraction derived from the enamel layer of developing porcine teeth. The enamel matrix derivative (EMD) is a gel-like material that consists of enamel matrix proteins, water and propylene glycol alginate, which is used as a carrier. The major enamel matrix proteins in EMD are amelogenins (90%). EMD also contains other proteins such as enamelin, ameloblastin, amelotin and various proteinases in a very low percentage. EMD plays a significant role in wound healing, promoting the formation of new blood vessels as well as collagen fibers in the connective tissue. It also promotes regeneration through the increase in cell attachment, the proliferation of periodontal ligament-fibroblasts and the increase in the expression of growth factors, molecules involved in osteogenesis and molecules involved in the regulation of bone remodeling [23,24].

Blood derivatives 

Blood derivatives are materials obtained by the patient’s own blood. These materials are not meant to create a mechanical barrier or a stabilization of the blood clot, but rather to induce regeneration by means of a potent production of growth factors by the platelets and other components of the blood clot obtained by the patient’s own blood. 

Platelet-rich plasma (PRP) was the first generation of blood derivatives, obtained by two cycles of centrifugation and characterized by a short-term release (1–8 days) of growth factors. Within 10 min, 70% of growth factors are already released, and within the first hour, almost 100% are released [25,26,27]. Nowadays the PRP has been replaced by platelet-rich fibrin (PRF), which needs a much simpler preparation without any use of anticoagulants. PRF provides a more stable material with a higher concentration of growth factors, as platelet-derived growth factor aa (PDGFaa), PDGFbb, PDGFab, transforming growth factor beta1 (TGF-b1), TGF-b2, vascular endothelial growth factor and epithelial growth factor. Additionally, T-lymphocytes, B-lymphocytes and monocytes are found within the first 25–30% proximal part of the clot and increase anti-bacterial and angiogenetic properties [28,29,30].

Blood derivatives have the advantage of being an attractive alternative option for patients who do not accept materials from allogenic or xenogeneic origin and at the same time reduce the risks of possible foreign body reactions. However, retrieving blood can cause additional local pain and discomfort and it may not be suitable in patients with a high hemorrhagic diathesis. Moreover, additional training is needed for clinical staff to learn the blood sampling procedure and the preparation of the material. 

In the literature, there is a great variety of clinical studies and few meta-analyses that have investigated the effect of the above-described materials, alone or combined, in the regeneration of molars with affected furcation sites, but the majority of them focus on a limited amount of materials. One recent review presented meta-analyses of different materials, but it did not include blood derivatives [31]. Therefore, the aim of this review is to propose an overview of the clinical effects of different regenerative materials when applied in periodontal surgical regeneration of molars affected by class II furcation defects.

## 2. Materials and Methods

For the present overview, a protocol was set up as close as possible to a systematic review. For the same reason, it is reported according to the PRISMA guidelines [32].

Focused Question

In periodontitis patients with class II furcation involvement, what are the clinical outcomes of different regenerative materials when compared to the surgical treatment with open flap debridement?

Eligibility Criteria

Following the PICOS strategy, only randomized control trials (S) considering periodontal patients with at least one molar with a class II furcation involvement (P) treated with regenerative materials (I) in comparison to OFD as control treatment (C) and a minimum follow-up period of 6 months were included. The studies should have also provided at least one of the following clinical outcomes (O) at the furcation sites: vertical clinical attachment level (VCAL), horizontal clinical attachment level (HCAL), horizontal probing depth (HPD) or vertical probing depth (VPD). 

The inclusion and exclusion criteria were defined prior to the beginning of the present study. Studies were considered to be eligible if they met the following inclusion criteria:−Randomized control trials (RCTs) in periodontitis patients with at least one class II furcation involvement were included. No distinction between maxillary and mandibular molars was considered.−Only RCTs that considered one or more of the above-mentioned treatment methods with open flap debridement (OFD) as control and a follow-up period of at least 6 months were included.−Studies published in English.

The exclusion criteria were:−Systematic reviews and meta-analyses, case series or case reports, narrative revisions and RCTs without OFD as control group.−The measurements of the considered outcomes were performed solely during the surgical phase and not clinically prior to the surgery.−Histological analyses, in vitro studies and trials conducted on animals.

Search Strategy

A thorough search of three different electronic bibliographic databases was performed without any language or date restrictions. The research on Medline-PubMed, Embase, Cochrane, Scopus and Web of Science was conducted up to 25 January 2022. Gray literature was searched in the OpenGrey database. The strategy plan consisted of at least two queries per electronic library which included phrases, medical subject heading terms, text words and combinations of them, as reported in the figure below (Figure 1).

Screening and Data extraction

The results collected after the database search were screened in three phases: at the title and preview, abstract and full text levels. Initially, after the title and abstract review, duplicates of articles were identified and removed, and afterwards, we proceeded with the full text review. Whenever a clear selection from the abstract was not possible to perform, full texts were then accessed. The titles and abstracts were screened independently by two reviewers (G.G. and F.T.). The articles that fulfilled all the eligibility criteria after reading the full text were included. Any disagreement between the two authors was resolved by discussion and consensus. If no agreement could be reached, a third author (S.B.) was requested to judge.

A quality assessment instrument (QAI) was developed based on Cochrane guidelines to objectively determine the risk of bias of the included RCTs. The QAI was composed of 7 stringent criteria and a scoring system in order to evaluate the methodological quality of the papers. Each study earned one point if the answer to the corresponding criteria was positive, otherwise no points if the answer was negative or unclear. The seven criteria were the following: random allocation, defined inclusion/exclusion criteria, blinding to patient and examiners, balanced experimental groups, identical treatment between groups (except for the intervention) and reporting of follow-up time. Low risk of bias was attributed to the studies that met seven of seven criteria while moderate and high risk were assigned to six of seven points and less than five points, respectively. The intervention of the third experienced reviewer (S.B.) was requested in case of any disagreement between the two authors. A PRISMA flowchart diagram was developed to show the selection process (Figure 2). 

## 3. Results

### 3.1. Study Selection

The Embase, Cochrane, Scopus, Web of Science and PubMed search identified in total 2064 articles, with 488 collected from PubMed, 765 from Embase, 305 from Cochrane, 260 from Scopus and 246 from Web of Science. One article was retrieved through the additional manual search. Furthermore, after checking 1300 papers as duplicates, 764 articles remained to be screened. After screening of titles and abstracts, 699 articles were excluded as they did not meet the inclusion criteria. From the remaining 65 publications, five articles were excluded due to some inconsistencies in data with the agreement of all the authors: Caton et al. [33] was excluded since the data were provided only in charts but not as numerical values. The same reason was considered for Dubrez et al. [34] at least for the primary outcome (VCAL); Lekovic et al. [35] reported mean ± standard error and not standard deviation; both Wang et al. [36] and Jaiswal et al. [37] reported unreliable data. Finally, 25 articles fulfilled the inclusion criteria and were included in this review [38,39,40,41,42,43,44,45,46,47,48,49,50,51,52,53,54,55,56,57,58,59,60,61,62]. Two of the included studies reported HPD measurements which were collected after the flap was opened [44,46]. The above-mentioned measurements were not considered in the following analysis according to the inclusion criteria.

### 3.2. Risk of Bias across the Included Studies

The quality assessment of the twenty-five studies was performed, with the QAI revealing that 13 papers were considered at high risk of bias, 6 at moderate risk and only 6 at low risk (Table 1).

Data from the outcomes of the studies were extracted and organized in tables. The studies were categorized in four groups based on the typology of regenerative materials used: non-absorbable membranes, absorbable membranes, blood derivatives and miscellaneous materials. The latter group consisted of studies where more than one type of materials was used and one study with enamel matrix derivatives (EMD), which did not fall into the other previous categories. VCAL was chosen as the primary outcome while VPD, VCAL and HCAL were used as secondary outcomes. 

Table 2 shows the population characteristics and the design of the studies. Table 3, Table 4, Table 5, Table 6 and Table 7D present the mean ± standard deviations (SD) of the outcome variables and their changes at baseline and at the follow-up of 6 months.

### 3.3. Qualitative Synthesis

#### 3.3.1. Non-Absorbable Membranes

All five studies identified using non-absorbable membranes (Table 2) were designed as split-mouth studies [38,39,40,41,42]. In four of these studies, patients were diagnosed with moderate to advanced periodontitis [39,40,41,42], while one of them did not specify the diagnosis of periodontitis [38]. In two studies, no patients with systemic diseases were included [38,40]. In the rest of them, the systemic conditions of the patients were not mentioned [39,41,42]. One study [40] specified that the population included did not use any antibiotics 3 months prior to the study, and none of the studies specified if smokers were included or not. Two studies mentioned that patients achieved a plaque index of 10% or lower [38,39]; in one study, the population had a 0.4–0.7 plaque score [42]; and in two studies, the oral hygiene of the population was not reported [40,41] (Table 2).

One study used an e-PTFE membrane [39], two studies used a PTFE membrane [38,41], one study an e-PTFE membrane with or without an antibiotic [48] and one study used an e-PTFE membrane and a coronally positioned flap [42] (Table 2).

The study that used only an e-PTFE membrane applied the membrane in 17 subjects with split-mouth design [39]. The study showed no statistically significant changes from baseline to the follow-up period for VCAL, VPD and HPD [39] (Table 3 and Table 7A).

The studies that used PTFE as an absorbable membrane recruited 21 [41] and 8 [38] patients, both in a split-mouth design. Both studies provided data based on the location of the furcation (buccal or lingual) where the treatment was applied (Table 3 and Table 7A). The buccal furcations in one study showed statistically significant changes in VCAL, VPD and HCAL for the test and only in VCAL and VPD changes in the control group [41]. The other study showed statistically greater changes in VCAL (2 ± 0.63 vs. 0.5 ± 0.42 mm) and VPD (2.88 ± 0.48 vs. 1.38 ± 0.65 mm) than the control group for the buccal furcations [38]. The lingual furcations in the first study showed statistically significant changes in VCAL, VPD and HCAL for the test group and in VPD and HCAL for the control group [41]. The other study presented significantly greater changes in VCAL (1.5 ± 0.46 vs. 0.13 ± 0.48 mm) and VPD (2.88 ± 0.55 vs. 1.25 ± 0.53 mm) for the lingual furcations [38]. The changes in VCAL and VPD were not of the same magnitude between the studies either for the buccal nor for the lingual furcations (Table 3 and Table 7A).

The study where an e-PTFE membrane was used with or without an antibiotic considered five patients in a split-mouth design [48]. The changes in VCAL and VPD were not statistically significant, and they were also similar among all treatment groups. On the other hand, HPD changes were not statistically significant. Moreover, the e-PTFE group resulted in two open furcations and the e-PTFE + antibiotic group resulted in one open furcation in contrast to the control groups, where no open furcation sites were found (Table 3 and Table 7A).

In the study where an e-PTFE membrane was used together with a coronally positioned flap, 28 patients were analyzed in a split-mouth design [42]. The results were provided separately for each furcation location and were presented in charts. Sufficient data were provided in the paper only for the mesial furcations. Based on that, only the mesial furcations were considered for the analysis. The changes in VCAL were statistically significant only for the test group (0.7 mm) but not for the control group (0.1 mm). The VPD changes were statistically significant for both test (1.6 mm) and control groups (1.3 mm) (Table 3 and Table 7A).

#### 3.3.2. Absorbable Membranes

The search identified four studies where absorbable membranes were used (Table 2); they were all split-mouth studies [43,44,45,46]. In two studies [43,44], patients were diagnosed with advanced periodontitis, whereas in the other two studies, this information was not specified [45,46]. Two out of the four studies included patients with no systemic conditions [44,45], one of them excluded patients who required antibiotic prophylaxis [43] and one of them did not report any specification [46]. Three studies did not include whether or not the population used any medication [43,44,45], and one of them included people who did not use any medication up to one month prior to the beginning of the study [45]. One study reported the smoking habit of the patients [43]. Two studies did not report the oral hygiene level of the patients [44,46], while two of them reported acceptable oral hygiene prior to the study based on gingival and plaque indexes [43,45].

The absorbable membranes used in one study were Vicryl mesh combined with a coronally positioned flap [43]. One study applied an autologous periosteal graft [45], and two studies applied collagen membranes [44,46]. 

The study that used the Vicryl mesh membrane included 11 subjects with the split-mouth design [43]. It showed a statistically significant improvement in VCAL and HPD at 6 months follow-up. In the comparison test vs. control, the test group showed a statistically significant gain in VCAL (2.18 ± 0.60 vs. 1.09 ± 0.94 mm) (Table 4 and Table 7B). 

The study that used autologous periosteal graft as an absorbable membrane included 12 subjects per treatment group [45], and both the test and control groups showed a statistically significant improvement in VCAL after therapy (Table 4 and Table 7B). When comparing Vicryl mesh and OFD, the study [45] showed a significantly greater gain in VCAL in the test group (2.17 ± 0.72 vs. 0.83 ± 0.72 mm) (Table 7B).

The studies where collagen membranes were applied included a range of 14–27 furcations per treatment group [44,46]. Both studies showed a statistically significant improvement in test groups for VPD and HPD, while neither showed an improvement in VCAL. When comparing test groups and controls, only one study [44] showed a statistically greater improvement in VPD (1.50 ± 0.76 vs. 0.86 ± 0.77 mm) (Table 4 and Table 7B). 

#### 3.3.3. Blood Derivatives

Six studies focusing on blood derivatives were identified (Table 2), three of which were parallel studies [47,48,49], two were split-mouth studies [50,51] and one was a split-mouth and parallel study [52]. Two included patients with chronic periodontitis [48,49], two included patients with chronic moderate to severe periodontitis [47,52] and two did not comment on the diagnosis of the periodontitis in their population [50,51]. In all six papers, no patients with underlying systemic conditions, using medication that can affect the periodontal therapy, or smoking were included. Furthermore, in four studies, the patients achieved acceptable oral hygiene prior to the study [48,49,50,51]; in one of them, the patients achieved a plaque score of 0.1–0.9 prior to the surgery [52]; and one did not comment on this topic [47] (Table 2).

The blood derivatives used were PRF or PRP in one study [48], one study applied PRF or PRF + 1% Alendronate [49], one study applied only PRF [51], one only PRP in the test group [50], one PRF or PRF + allograft [47] and one applied PRF or β-TCP in the test groups [52] (Table 2).

In the study where PRP and PRF were used, 12 and 13 patients were recruited, respectively, for the test groups and 12 patients for the control group according to a three-arm parallel study [48]. The changes in VCAL, VPD and HCAL were statistically significant in all groups. The VCAL changes were significantly greater for the PRF group (2.78 ± 0.85 mm) compared to the control group (1.37 ± 0.58 mm), and for the PRP group (2.71 ± 1.04 mm) compared to the control group. The between-groups comparisons of the VPD (4.29 ± 1.04, 3.92 ± 0.93 vs. 1.58 ± 1.02 mm) and HCAL (2.75 ± 0.94, 2.5 ± 0.83 vs. 1.08 ± 0.50 mm) were also similar (Table 5 and Table 7C).

One study applied PRF with or without 1% alendronate in a three-arm parallel study where 24 patients were considered in the PRF group, 25 in the PRF + 1% alendronate group and 23 in the control group [49]. The changes in VCAL, VPD and HCAL were statistically significant in all groups. The changes in VCAL were statistically greater for the PRF + 1% Alendronate (4.12 ± 0.6 mm) compared to PRF (3.39 ± 0.49 mm) and to the control group (2.33 ± 0.48). The change in VCAL was significantly greater for the PRF as compared to the control group. The between-groups comparisons for the changes in VPD (4.4 ± 0.57 vs. 3.69 ± 0.76 vs. 2.41 ± 0.77 mm) and HCAL (3.64 ± 0.90 vs. 2.86 ± 0.062 vs. 2.04 ± 0.35 mm) were similar (Table 5 and Table 7C).

In one study, PRF was applied in 18 patients according to a split-mouth design [51]. In both test and control groups, the changes in VCAL (2.333 ± 0.485 vs. 1.278 ± 0.461 mm), VPD (4.056 ± 0.416 vs. 2.889 ± 0.676 mm) and HCAL (2.667 ± 0.594 vs. 1.889 ± 0.758 mm) were statistically significant. For all the above measurements, the test group showed significantly greater changes (Table 5 and Table 7C).

PRP was applied in one study where 20 patients were used in a split-mouth design [50]. The VCAL changes for the test group (2.50 ± 1.64 mm) were statistically significant and also significantly greater than the control group (0.10 ± 1.10 mm), which resulted not statistically significant. Similar outcomes were presented for the VPD (2.31 ± 1.41 vs. 0.80 ± 1.31 mm) and HCAL (2.5 ± 1.17 vs. 0.8 ± 0.63 mm) (Table 5 and Table 7C).

PRF or PRF + allograft was applied in one study where 20 patients were used per group in a three-arm parallel study [47]. The VCAL, VPD and HPD changes were statistically significant in each group. The changes in VCAL were statistically greater for both the PRF group (3.55 ± 1.05 mm) and the PRF + allograft group (3.90 ± 0.72 mm) when compared to the control group (1.35 ± 0.49 mm). The changes in VPD (3.80 ± 0.77, 4 ± 0.79 mm) were also significantly greater for both groups compared to the OFD group (1.50 ± 0.76 mm). The HPD changes were similar among the groups. Furthermore, the changes in VCAL and VPD were similar between the two test groups, with the PRF + allograft group presenting slightly better results (Table 5 and Table 7C).

In the study where PRF or β-TCP were applied, 31 patients were enrolled in a split-mouth parallel study [52]. The VCAL, VPD and HCAL changes were statistically significant among all the groups. The VCAL changes for the PRF group (2.40 ± 0.91 mm) and the β-TCP group (2.53 ± 0.83 mm) were statistically greater than the OFD group (0.93 ± 0.46 mm). Similar results were also presented for the HCAL changes (2.40 ± 1.06, 2.27 ± 0.46 vs. 0.73 ± 0.46 mm). Neither test group presented statistically greater results than the OFD for the VPD changes. The comparison of the changes in VCAL, VPD and HCAL between the test groups was not statistically significant, but the outcomes were similar (Table 5 and Table 7C).

In the studies where PRF was used as a test group, the changes in VCAL and VPD were split in two results. Two of the studies [47,49] presented slightly superior results for those changes than the other three [48,51,52]. The changes in HCAL were similar in four of the studies [48,49,51,52], and one of them did not measure HCAL [47] (Table 5 and Table 7C).

In the studies where PRP was used as a test group, the changes in VCAL and HCAL were similar [48,50] (Table 5 and Table 7C).

#### 3.3.4. Miscellaneous Materials

The search also identified 10 studies where combinations of materials were applied. The materials that did not fall into any of the other previously described categories were also added to this group [53,54,55,56,57,58,59,60,61,62]. Five of these studies had a parallel design [54,59,60,61,62], four had a spilt-mouth design [53,55,57,58] and one was a combination of a split-mouth and a parallel design [56]. In one of these studies, patients were diagnosed with chronic periodontitis [55], one specified inclusion of patients with chronic but not aggressive periodontitis [59], two with advanced chronic periodontitis [56,60], one with moderate chronic periodontitis [54], one with stages 3–4 periodontitis [61], two with moderate to advanced periodontitis [53,58] and two did not report any diagnosis [57,62]. None of the studies included patients with systemic conditions except one, which did not add any specific comment [56]. None of the studies included people using medications chronically or patients who used antibiotics prior to the commencement of the studies, except for one study which provided no data [56]. Five studies excluded smoking patients [54,55,59,60,61], four studies provided no data on smoking [53,56,58,62] and one study included both smokers and non-smokers [57]. Four studies reported that the population had acceptable oral hygiene prior to the study [57,59,60,62]; in two studies, the population achieved ≤20% plaque index or plaque score prior to the surgery [56,61]; in one study, patients achieved a plaque score 10% or lower [53]; and three studies did not comment on the oral hygiene of the population [54,55,58] (Table 2).

In one study, an allograft with or without a collagen membrane was used in a three-arm parallel study considering nine patients per group [62]. A significant statistical improvement after therapy for VPD in the allograft group and for HPD in the allograft + collagen membrane group was recorded. No statistically significant differences were reported for VCAL, VPD and HPD between the groups (Table 6 and Table 7D).

In the study where PRF and synthetic graft were used with or without rosuvastatin (RSV), 35 patients were involved in a parallel study [59]. A statistically significant improvement in VCAL, VPD and HCAL in all groups was obtained. The VCAL changes were significantly greater for the RSV + PRF + synthetic graft (4.17 ± 0.70 mm) in comparison with the PRF + synthetic graft group (3.31 ± 0.52 mm). Both were statistically greater than the control group (1.82 ± 0.78 mm). Similar results were also shown for the changes in VPD and HCAL between the groups (Table 6 and Table 7D).

PRP and xenograft with a collagen membrane were used in a study where 26 patients were enrolled in a split-mouth design [57]. The improvements after therapy for VCAL and VPD were statistically significant for both the test and control groups. The changes for VCAL were statistically greater in the test group (3.29 ± 0.42 vs1.68 ± 0.31 mm) and similar for VPD (4.07 ± 0.33 vs. 2.49 ± 0.38 mm) (Table 6 and Table 7D).

In one study, a combination of composite graft, a PTFE membrane and a coronally positioned flap was tested, and 30 patients were included in a parallel study [60]. As shown in Table 6, the changes in VCAL, VPD and HCAL were statistically significant for both the test and control groups. The changes between the groups in VCAL (3.05 ± 0.6 vs. 0.65 ± 0.6 mm), VPD (3.56 ± 0.6 vs. 0.6 ± 1 mm) and HCAL (3.45 ± 1.3 vs. 0.55 ± 0.7 mm) were statistically significant in favor of the test group.

In the study where a synthetic graft together with peptide P-15 and a coronally positioned flap were applied, 12 patients participated in a split-mouth study [55]. The differences between the test and control groups were not statistically significant for any of the parameter studied. The HCAL improvement after therapy was statistically significant only for the control group while the improvement in VCAL was statistically significant in both groups.

One study used a xenograft together with a collagen membrane in a mixed parallel and split-mouth study where 16 patients were enrolled in the test group and 11 in the control group [56]. The changes after therapy in VCAL, VPD and HCAL were statistically significant only for the test group. For the comparison between groups, the changes in VCAL (1.8 ± 1.8 vs. 0.6 ± 2.06 mm), VPD (2 ± 1.7 vs. 0.3 ± 1.37 mm) and HCAL (2.2 ± 2.2 vs. −0.2 ± 1.6 mm) were significantly greater for the test group (Table 6 and Table 7D).

The search included only one study applying EMD where 10 patients per group were included in a two-arm parallel study [54]. The changes in VCAL, VPD and HCAL from the baseline were statistically significant for both groups. However, the test group showed a statistically significant greater improvement in HPD (1.9 vs. 0.6 mm, SD not reported) but not in VCAL and VPD (Table 6 and Table 7D).

One study applied an autologous graft with or without the adjunct of L-PRF in a three-arm parallel study where 18 patients per group were included [61]. Statistically significant improvements after therapy were reported for HCAL, VCAL and PPD for all three groups. The VCAL change was significantly greater for the L-PRF + autologous bone graft group (2.139 ± 0.278 mm) when compared to the autologous bone graft alone group (1.994 ± 0.276 mm), and both were statistically greater than the control group (0.811 ± 0.284 mm). Similar statistical results were shown among the three groups for the changes in VPD (2.52 ± 0.71 vs. 2.15 ± 0.17 vs. 1.00 ± 0.71 mm) and HCAL (2.30 ± 0.18 vs. 1.61 ± 0.18 vs. 0.87 ± 0.18 mm) (Table 6 and Table 7D).

In one study, the application of bioactive glass was investigated in a split-mouth study where 15 patients were included [53]. In this study, the mean VCAL measurements and the standard error were provided. The changes in VCAL were statistically greater for the test group (3.27 ± 0.27 mm) compared to the control group (2.40 ± 0.24 mm) (Table 6 and Table 7D).

One study used a synthetic graft together with PRP in a split-mouth study where 11 patients were enrolled [58]. There was a statistically significant change in VCAL and VPD for both groups, but the numerical changes were only presented in figures. Furthermore, the test group presented a statistically greater mean change in HPD (2.3 mm) as compared to the control group (1.7 mm) (Table 6 and Table 7D).

## 4. Discussion

The aim of the current review was to provide an overview of the clinical effect of the use of different regenerative materials, alone or combined, in the surgical treatment of class II furcation defects in patients with periodontitis. In our search, only RCTs with OFD as control group were included. In a previous meta-analysis, OFD has shown to provide clinical improvement, although limited, in mandibular class II furcations [63] with relatively low cost and morbidity. The question is whether the additional costs for the adjunct of regeneration materials are justified by the additional clinical improvements. 

The results from the studies included in the present review suggest that surgical therapy combined with regenerative materials can lead to mildly to moderately better clinical outcomes, particularly for VCAL and VPD, in mandibular buccal class II furcations when compared to OFD alone. Less consistent results have been reported for maxillary and lingual mandibular class II furcation defects. No conclusions can be made for class III furcations because of the very limited data available. 

These conclusions are supported by a number of meta-analyses already available in the literature, which assessed the use of blood derivatives, EMD, absorbable and non-absorbable membranes and bone graft substitutes [31,64,65,66,67,68]. More specifically, the meta-analysis by Jepsen et al. [31] is one of the most complete available. In their Bayesan network analyses, the authors showed a mean treatment improvement of a 1.6 mm gain in HCAL and a 1.3 mm reduction in VPD and VCAL in comparison with OFD. In addition, the authors suggested that the treatments with a bone graft alone or combined with an absorbable barrier seemed to show the highest chance of achieving the treatment outcome for mandibular class II furcations, followed by the use of EMD alone. On the other side, the authors reported higher incidence of post-operative complications when barriers were used, particularly non-absorbable, in comparison with EMD. This can be due not only to the biological characteristics of the materials, but also to the differences in the need for high technical skills required in order to apply barriers in comparison with EMD. In our review, one study which used EMD combined with an absorbable barrier and bone graft was included and it showed some better results than the combination of the biomaterials alone or the OFD. However, the results of this RCT are not corroborated by a recent meta-analysis, which could not show statistically significant differences between EMD alone or combined with a bone substitute [65]. 

Although very extensive, the review of Jepsen et al. did not include blood derivatives [31]. These materials have been introduced in relatively more recent times in comparison with the others. In the past 15 years, the enthusiasm about the clinical and histological potential of blood derivatives elicited the initiation of several RCTs in different oral applications [69]. The RCTs included in the current review also showed the highest quality assessment when compared with the other trials. A meta-analysis by Troiano et al. [68] analyzed the additional effect of blood derivatives as the only material in comparison to OFD. The authors reported an additional improvement of 1.8 mm for VPD, 1.5 mm for VCAL and 1.4 mm for HCAL. However, the authors noticed that the three studies included in the meta-analysis were all performed in the same country. This may make an extrapolation of the results to other cultural situations more difficult. A more recent meta-analysis from Tarallo et al. [67] confirmed these findings and further investigated the possible additional effect of the combination of blood derivatives with bone graft materials. They concluded that the additional use of a bone graft yielded only an additional statistically significant improvement of 0.7 mm in VCAL in comparison with the blood derivative alone. No patient-centered outcomes were reported by any of the studies included. 

The current review aimed to include a wide palette of treatment options. However, a meta-analysis to compare all materials was not implemented. The main reasons were the large heterogeneity of the study populations, the differences in the study designs, treatment protocols and primary outcomes, the low to moderate quality of the majority of the studies resulting from the quality assessment (the main exceptions were the studies which used blood derivatives) and the scarce availability of multi-armed randomized control trials that compare different materials with each other and with the OFD. In addition, the majority of studies have a small sample size, a relatively short follow-up (6 to 12 months) and no reported long-term data on the stability of the clinical results or tooth loss. These limitations are also often reported the other previous reviews. 

With the available scientific evidence, the choice of a single specific material or a combination of multiple materials still remains a challenge. Although the use of a bone graft alone or combined with other materials seems to be more promising, in clinical settings, other parameters should be considered rather than the mere mean data reported. In the available trials, there is a lack of detailed analyses of the anatomy and morphology of the bone defects. These factors may play a role in the final clinical outcome, and they can be important to assess the choice of the most compatible material in terms of biological and mechanic characteristics. Future RCTs should record in detail the anatomical configurations of the furcations and of the relative bony defects, and these data should be taken into account in the final statistical analyses of the clinical results. In addition, there is a lack of information about the quality of the regeneration achievable in the furcation areas with the different materials. The non-absorbable membranes offer the opportunity to assess the bone growth at the re-entry surgery, necessary to remove the barrier. This would not be the case for the other materials. In future trials, assessing the amount of regeneration through histological or radiographic analyses would be suitable, but it can be ethically challenging. Alternatively, investigators may consider a clinical assessment of the change in the bone level through the bone sounding as a relatively non-invasive surrogate measure of bone healing. 

Another important point of discussion is the limited availability of patient-centered outcomes, e.g., pain experienced during the post-operative healing, clinical complications, patient’s satisfaction and acceptance of the treatment. Moreover, the costs of the use of the xenografts, allografts and alloplastic materials, and the invasiveness of the procedures required to harvest autologous materials, together with the risks of post-surgical complications and morbidity should be weighted with clinical significance of the benefit. All these factors should be critically evaluated in every specific clinical situation in order to assess the indication of the use of the regenerative materials in comparison with other treatment modalities.

## 5. Conclusions

The use of regenerative materials may yield a mild to moderate superior clinical improvement in mandibular class II furcation defects on OFD alone. However, the scarcity of multi-armed trials does not allow us at the moment to objectively assess the superiority of one single material or a specific combination of materials and help the clinician in this choice. Previous meta-analyses suggest that the combination of a bone graft with either barrier or blood derivatives may yield the highest clinical improvement. However, the lack of patient-centered outcomes does not allow us to weigh the risks of complications and morbidities with the significant clinical benefits. Therefore, regeneration in class II furcations cannot be advised routinely. This choice remains based on the clinician’s experience, the costs and the acceptance of the patient. For future research, there is a need to standardized protocols, surgical procedures and the assessment of the primary clinical outcomes. Furthermore, more data on cost-effectiveness, post-surgical patient-centered outcomes and more long-term data on the clinical stability of the results and tooth survival are necessary in order to objectively justify the choice of the regenerative procedure on solely open flap debridement. These difficulties can be partly overcome by setting up sufficiently powered trials, either monocentric or multi-center, with homogeneous protocols which will allow multi-level analyses that will take into account as many factors as possible to deliver personalized treatment and advice for specific patients’ groups.

## Figures and Tables

**Figure 1 materials-15-03194-f001:**
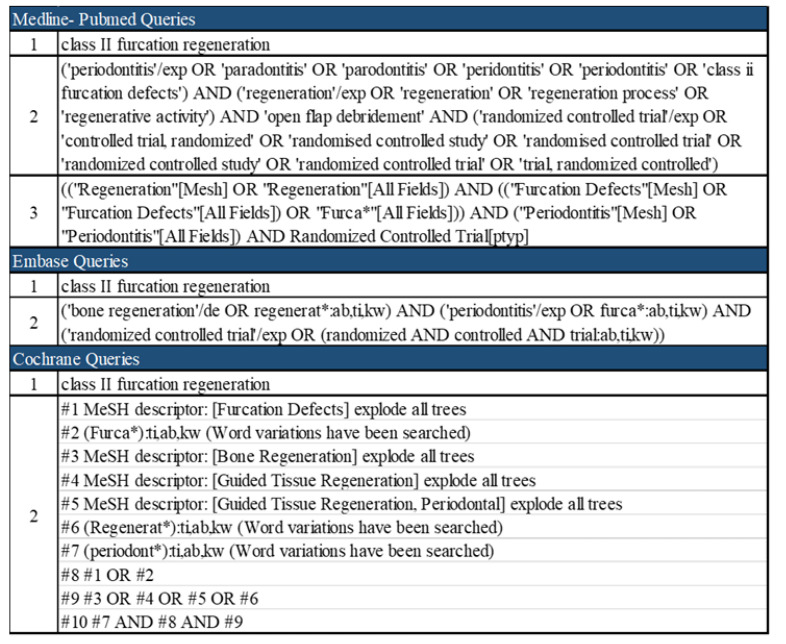
Queries and MESH terms used on three databases.

**Figure 2 materials-15-03194-f002:**
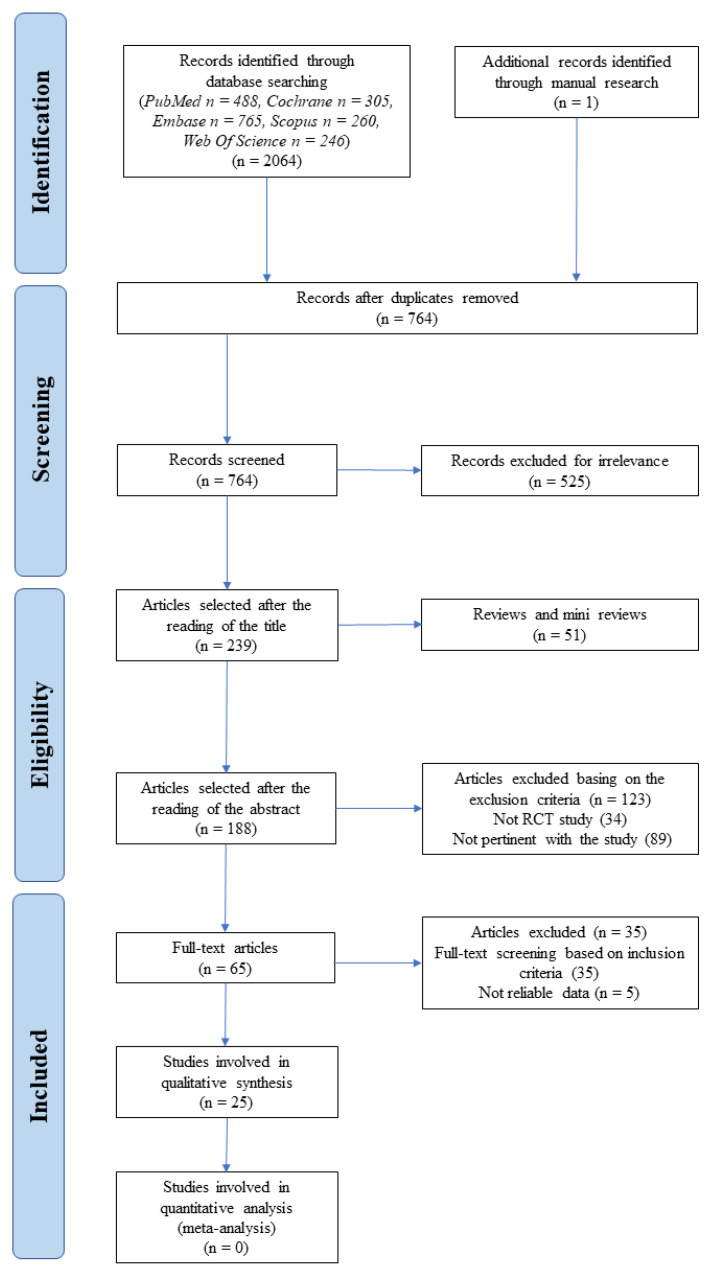
PRISMA flowchart for studies selection. The reasons for excluding articles were the following: non-RCT studies, no furcation defects, no OFD as control group, animal and in vitro studies, irrelevant subjects, no periodontal patients and/or regeneration, peri-implant treatment, short follow-up period, unreliable data and language.

**Table 1 materials-15-03194-t001:** Risk of bias summary: review authors’ judgement about each risk of bias item presented across all included RCTs.

Author (Year)	Random Allocation	Inclusion/Exclusion Criteria Clearly Defined	Blinding of Participants	Blinding of Examiners	Balanced Experimental Groups	Identical Treatment between the Groups	Reporting of Follow-Up	Total
**Non-Resorbable Membranes**								
Avera et al. (1998)	Y	Y	N	Unclear	Y	Y	Y	**5 of 7**
Metzler et al. (1991)	Y	N	N	N	Y	Y	Y	**4 of 7**
Mombelli et al. (1996)	Y	Y	Y	Y	Y	Y	Y	**7 of 7**
Pontoriero et al. (1988)	Y	Y	N	Unclear	Y	Y	Y	**5 of 7**
Pontoriero et al. (1995)	Y	N	N	N	Y	Y	Y	**4 of 7**
**Absorbable Membranes**								
Balusubramanya et al. (2012)	Y	Y	Unclear	Unclear	Y	Y	Y	**5 of 7**
Paul et al. (1992)	Y	N	N	N	Y	Y	Y	**5 of 7**
Verma et al. (2011)	Y	Y	Unclear	N	Y	Y	Y	**5 of 7**
Yukna et al. (1996)	Y	N	Unclear	Y	Y	Y	Y	**5 of 7**
**Blood Derivates (PRP, PRF)**								
Agarwal et al. (2020)	Y	Y	Unclear	Y	Y	Y	Y	**6 of 7**
Bajaj et al. (2013)	Y	Y	Y	Y	Y	Y	Y	**7 of 7**
Kanoriya et al. (2017)	Y	Y	Y	Y	Y	Y	Y	**7 of 7**
Pradeep et al. (2009)	Y	Y	Y	Y	Y	Y	Y	**7 of 7**
Sharma et al. (2011)	Y	Y	Y	Y	Y	Y	Y	**7 of 7**
Siddiqui et al. (2016)	Y	Y	Unclear	Unclear	Y	Y	Y	**5 of 7**
**Miscellaneous Materials**								
Anderegg et al. (1999)	Y	Unclear	Unclear	N	Y	Y	Y	**4 of 7**
Chitsazi et al. (2007)	Y	Y	Unclear	N	Y	Y	Y	**5 of 7**
Eto et al. (2007)	Y	Y	Unclear	Y	Y	Y	Y	**6 of 7**
Houser et al. (2001)	Y	Unclear	Unclear	Y	N	Y	Y	**4 of 7**
Lekovic et al. (2003)	Y	Y	Unclear	Y	Y	Y	Y	**6 of 7**
Mohamed et al. (2016)	N	Unclear	Unclear	N	Y	Y	Y	**3 of 7**
Pradeep et al. (2016)	Y	Y	Y	Y	Y	Y	Y	**7 of 7**
Santana et al. (2009)	Y	Y	Unclear	Y	Y	Y	Y	**6 of 7**
Serroni et al. (2021)	Y	Y	Unclear	Y	Y	Y	Y	**6 of 7**
Tsao et al. (2006)	Y	Y	Unclear	Y	Y	Y	Y	**6 of 7**

RTC, randomized clinical trials; PRP, platelet-rich plasma; PRF, platelet-rich fibrin. One point was given when the answer to the bias item was “Yes”. If the answer was “No” or “Unclear”, no more points were given.

**Table 2 materials-15-03194-t002:** Study design of the included studies.

Author (Year)	Study Design	Age	Sex	Diagnosis of Periodontitis	Systemic Conditions	Medication	Smoking	Oral Hygiene
**Non-Absorbable Membranes**								
Mombelli et al. (1996)	split mouth	35–65	NR	moderate to advanced	NO	no antibiotics in the past 3 months	NR	NR
Pontoriero et al. (1988)	split mouth	22–65	NR	moderately advanced to advanced	NR	NR	NR	NR
Metzler et al. (1991)	split mouth	29–64	13 M 4 F	moderate to advanced adult	NR	NR	NR	plaque score 10% or lower
Avera et al. (1998)	split mouth	mean age: 42 ± 6.5	3 M 5 F	NR	NO	NR	NR	plaque score 10% or lower
Pontoriero et al. (1995)	split mouth	21–59	NR	moderately advanced to advanced	NR	NR	NR	0.4–0.7 plaque score
**Absorbable Membranes**								
Balusubramanya et al. (2012)	split mouth	18–60	NR	advanced	no patients requiring antibiotic prophylaxis	NR	NO	acceptable
Verma et al. (2011)	split mouth	28–49	7 M 5 F	NR	NO	not prior to 1 month	NR	acceptable
Paul et al. (1992)	split mouth	42–65	6 M 1 F	advanced	NO	NR	NR	NR
Yukna et al. (1996)	split mouth	46.8	15 M 12 F	NR	NR	NR	NR	NR
**Blood Derivatives**								
Bajaj et al. (2013)	parallel arms	mean age: 39.4	22 M 20 F	chronic	NO	no medication affecting periodontal healing	NO	acceptable
Kanoriya et al. (2017)	parallel arms	30–50	36 M 36 F	chronic (not aggressive)	NO	no medication affecting periodontal therapy	NO	acceptable
Sharma et al. (2011)	split mouth	mean age: 34.2	10 M 8 F	NR	NO	no medication affecting periodontal healing	NO	acceptable
Pradeep et al. (2009)	split mouth	mean age: 42.8	10 F 10 M	NR	NO	no medication affecting wound healing	NO	acceptable
Agarwal et al. (2020)	parallel arms	30–65 mean age: 46 ± 15	20 M 26 F	chronic moderate to severe	NO	NO	NO	NR
Siddiqui et al. (2016)	split mouth/parallel arms	30–50	24 M 7 F	chronic moderate to severe	NO	no antibiotics or medications affecting the periodontal therapy 6 months prior to the study	NO	Plaque score: 0.1–0.9
**Miscellaneous Materials**								
Tsao et al. (2006)	parallel arms	mean age: 54.4 ± 9.8	15 M 12 F	NR	NO	#no steroids (only for topical use)/not for 1 month #no antibiotics within 3 months prior to enrollment #not chronically treated with medication affecting periodontal status	NR	acceptable
Pradeep et al. (2016)	parallel arms	25–55	60 M 50 F	chronic (not aggressive)	NO	#no antibiotics in the preceding 6 months #no drugs affecting periodontal wound healing	NO	acceptable
Lekovic et al. (2003)	split mouth	mean age: 38 ± 11	12 M 14 F	NR	NO	no medication causing gingival enlargement	9 smokers 17 non-smokers	acceptable
Santana et al. (2009)	parallel arms	41–63 mean age: 48.3	26 M 34 F	advanced chronic (not aggressive)	NO	not for 6 months/no	NO	acceptable
Eto et al. (2007)	split mouth	34–63 mean age: 44.3	NR	chronic	NO	no medication affecting periodontal healing	NO	NR
Houser et al. (2001)	split mouth/parallel arms	mean age: 46	13 M 8 F	advanced adult	NR	NR	NR	20% plaque index prior to the surgical therapy
Chitsazi et al. (2007)	parallel arms	32–48 mean age: 40	7 M 3 F	chronic moderate to severe	NO	no antibiotics in the past 6 months	NO	NR
Serroni et al. (2021)	parallel arms	39–65 mean age: 54 ± 14	22 M 22 F	stage 3–stage 4	NO	no medication affecting periodontal healing 6 months prior to the study	NO	Full mouth plaque score < 20%
Anderegg et al. (1999)	split mouth	42–67 Mean age: 55	9 M 6 F	moderate to advanced	NO	no medication at least 6 months prior to the study	NR	Plaque index ≤ 10%
Mohamed et al. (2016)	split mouth	38–52	14 M	moderate to severe	NO	No medication 6 months prior to the study	NR	NR

M: male, F: female, NO: patients with systemic condition were excluded, NR: not reported.

**Table 3 materials-15-03194-t003:** Results reported at baseline and at follow-up in papers where non-resorbable membranes were used.

Study (Year)	Intervention	N	Follow-Up Period (Months)	Outcomes (pre-op and post-op)			
				VCAL (mm)	VPD (mm)	HCAL (mm)	HPD (mm)
**Non-Resorbable Membranes**							
Mombelli et al. (1996)	E-PTFE	5 furc	12.5	-	3.6 ± 1.52 NR	-	-
	E-PTFE + antibiotic	5 furc	12.5	-	3.8 ± 0.84 NR	-	-
	OFD + antibiotic	5 furc	12.5	-	4.2 ± 1.09 NR	-	-
	OFD	5 furc	12.5	-	3.4 ± 1.14 NR	-	-
Pontoriero et al. (1988) (A)	PTFE	21 subj/furc (buccal)	6	7.3 ± 1.1 3.2 ± 1.4	6 ± 0.9 1.5 ± 1.2	4.4 ± 1.2 0.3 ± 0.4	-
	OFD	21 subj/furc (buccal)	6	7.3 ± 1.5 5.8 ± 1.1	6 ± 1.6 3.2 ± 0.8	4 ± 0.8 2.0 ± 1.1	-
Pontoriero et al. (1988) (B)	PTFE	21 subj/furc (lingual)	6	7.5 ± 1.6 4.6 ± 1.7	6 ± 1 2.5 ± 1	4 ± 0.8 0.7 ± 1.0	-
	OFD	21 subj/furc (lingual)	6	7.2 ± 0.6 6.6 ± 0.7	5.4 ± 0.5 3.3 ± 0.5	4.4 ± 1.2 2.2 ± 1.2	-
Metzler et al. (1991)	E-PTFE	17 subj/furc	6	6.4 ± 1.3 5.4 ± 1.3	5.0 ± 1.5 3.3 ± 1.6	-	-
	OFD	17 subj/furc	6	5.7 ± 1.5 5.5 ± 1.7	4.6 ± 1.4 3.7 ± 1.5	-	-
Avera et al. (1998) (A)	PTFE	8 subj/furc (buccal)	9	-	7 ± 0.66 4.12 ± 0.6	-	-
	OFD	8 subj/furc (buccal)	9	-	6.25 ± 0.59 4.87 ± 0.57	-	-
Avera et al. (1998) (B)	PTFE	8 subj/furc (lingual)	9	-	6.63 ± 0.57 3.75 ± 0.59	-	-
	OFD	8 subj/furc (lingual)	9	-	5.75 ± 0.41 4.5 ± 0.45	-	-
Pontoriero et al. (1995)	E-PTFE + CPF	28 subj/furc (mesial)	6	7 6.3	5.7 4.1	-	-
	OFD	28 subj/furc (mesial)	6	7.2 7.1	5.6 4.3	-	-

VCAL: vertical clinical attachment level, VPD: vertical probing depth, HPD: horizontal probing depth, HCAL: horizontal clinical attachment level, OFD: open flap debridement, CPF: coronally positioned flap, A: buccal furcation, B: lingual furcation, subj: subjects, furc: furcations, NR: not reported, PTFE: Polytetrafluoroethylene, E-PTFE: expanded PTFE.

**Table 4 materials-15-03194-t004:** Results reported at baseline and at follow-up in papers where absorbable membranes were used.

Study (Year)	Intervention	N	Follow-Up Period (Months)	Outcome (pre-op and post-op)			
				VCAL (mm)	VPD (mm)	HCAL (mm)	HPD (mm)
**Absorbable Membranes**							
Balusubramanya et al. (2012)	v. m. + CPF	11 furc	6	4.09 ± 0.70 1.91 ± 0.70	-	-	8.27 ± 1.19 6.73 ± 0.90
	OFD	11 furc	6	3.82 ± 0.60 2.73 ± 1.01	-	-	7.73 ± 1.35 6.36 ± 1.12
Verma et al. (2011)	aut. periost. gr.	12 subj/furc	6	5.33 ± 0.49 3.17 ± 0.39	-	-	-
	OFD	12 subj/furc	6	5.50 ± 0.80 4.67 ± 0.78	-	-	-
Paul et al. (1992)	c. m.	7 subj/14 furc	6	6.86 ± 1.77	5.00 ± 0.93 NR	-	-
	OFD	7 subj/14 furc	6	5.79 ± 1.26	4.29 ± 0.59 NR	-	-
Yukna et al. (1996)	c. m.	27 furc	6–12	-	5.8 ± 1.2 4.1 ± 1.3	-	-
	OFD	27 furc	6–12	-	5.5 ± 1.6 4.2 ± 1.6	-	-

VCAL: vertical clinical attachment level, VPD: vertical probing depth, HPD: horizontal probing depth, HCAL: horizontal clinical attachment level, v. m.: vicryl mesh, CPF: coronally positioned flap, OFD: open flap debridement, aut. periost. gr.: autologous periosteal graft, c. m.: collagen membrane, subj: subjects, furc: furcations, NR: not reported.

**Table 5 materials-15-03194-t005:** Results reported at baseline and at follow-up in papers where blood derivatives were used.

Study (Year)	Intervention	N	Follow-Up Period (Months)	Outcome (pre-op and post-op)			
				VCAL (mm)	VPD (mm)	HCAL (mm)	HPD (mm)
**Blood Derivatives**							
Bajaj et al. (2013)	PRF	12 (24 furc)	9	7.42 ± 0.78 4.54 ± 0.51	7.29 ± 0.95 3.0 ± 0.51	8.17 ± 0.82 5.42 ± 0.72	-
	PRP	13 (25 furc)	9	7.08 ± 0.72 4.38 ± 0.71	7.17 ± 1.01 3.25 ± 0.68	8.08 ± 0.65 5.58 ± 0.72	-
	OFD	12 (23 furc)	9	7.32 ± 0.80 5.92 ± 0.70	6.87 ± 0.90 5.29 ± 0.99	7.96 ± 0.86 6.87 ± 0.85	-
Kanoriya et al. (2017)	PRF	24 subj/furc	9	7.56 ± 0.94 4.17 ± 0.83	7.73 ± 1.35 4.04 ± 0.87	7.13 ± 0.75 4.26 ± 0.81	-
	PRF + 1% ALN	25 subj/furc	9	7.52 ± 0.91 3.4 ± 0.57	7.52 ± 1.22 3.12 ± 0.88	7.16 ± 1.02 3.52 ± 0.65	-
	OFD	23 subj/furc	9	7.41 ± 0.92 5.08 ± 0.88	7.66 ± 1.27 5.25 ± 1.15	7.08 ± 0.82 5.04 ± 0.80	-
Sharma et al. (2011)	PRF	18 subj/furc	9	7.39 ± 1.145 5.06 ± 1.434	6.39 ± 1.145 2.33 ± 1.029	8.83 ± 1.618 6.17 ± 1.654	-
	OFD	18 subj/furc	9	7.33 ± 1.029 6.06 ± 1.162	6.33 ± 1.029 3.44 ± 1.042	8.94 ± 1.474 7.06 ± 1.349	-
Pradeep et al. (2009)	PRP	20 subj/furc	6	8.40 ± 1.71 6.40 ± 1.71	6.00 ± 0.94 3.70 ± 0.95	10.60 ± 2.07 8.10 ± 2.13	-
	OFD	20 subj/furc	6	7.00 ± 1.05 6.90 ± 1.66	5.10 ± 1.20 4.30 ± 1.64	8.70 ± 1.64 7.90 ± 1.85	-
Agarwal et al. (2020)	PRF	20 subj/furc	9	7.15 ± 0.67 3.60 ± 0.99	6.35 ± 0.93 2.55 ± 0.51	-	5.30 ± 0.66 3.50 ± 0.69
	PRF + al. gr.	20 subj/furc	9	7.15 ± 0.67 3.25 ± 0.44	6.30 ± 0.73 2.30 ± 0.47	-	5.20 ± 0.77 3.40 ± 0.59
	OFD	20 subj/furc	9	6.90 ± 0.64 5.55 ± 0.51	6.10 ± 0.85 4.60 ± 0.60	-	5.20 ± 0.62 3.85 ± 0.77
Siddiqui et al. (2016)	PRF	15 furc	6	5.47 ± 1.30 3.07 ± 1.03	3.73 ± 1.22 1.47 ± 0.64	4.60 ± 0.91 2.20 ± 0.86	-
	β-TCP	15 furc	6	5.53 ± 1.25 3 ± 0.85	4 ± 1.25 1.53 ± 0.52	4.53 ± 0.74 2.27 ± 0.59	-
	OFD	15 furc	6	5.47 ± 1.06 4.53 ± 1.13	4.07 ± 1.10 3.04 ± 0.77	4.47 ± 0.83 3.73 ± 0.80	-

VCAL: vertical clinical attachment level, VPD: vertical probing depth, HPD: horizontal probing depth, HCAL: horizontal clinical attachment level, PRF: platelet-rich fibrin, PRP: platelet-rich plasma, OFD: open flap debridement, ALN: alendronate, al. gr.; allograft, β-TCP: beta-tricalcium phosphate, subj: subjects, furc: furcations.

**Table 6 materials-15-03194-t006:** Results reported at baseline and at follow-up in papers where miscellaneous materials were used.

Study (Year)	Intervention	N	Follow-Up Period (Months)	Outcome (pre-op and post-op)			
				VCAL (mm)	VPD (mm)	HCAL (mm)	HPD (mm)
**Miscellaneous Materials**							
Tsao et al. (2006)	al. gr.	9 subj/furc	6	5.1 ± 2.8 5.0 ± 1.9	4.3 ± 1.8 3.4 ± 1.4	-	3.9 ± 1.1 2.7 ± 1.4
	al. gr. + c. m.	9 subj/furc	6	5.2 ± 2.0 5.6 ± 2.4	4.6 ± 1.1 3.9 ± 1.5	-	4.2 ± 1.2 3.1 ± 1.2
	OFD	9 subj/furc	6	5.4 ± 1.3 6.3 ± 1.8	4.7 ± 0.9 4.5 ± 0.9	-	4.7 ± 1.7 3.8 ± 1.6
Pradeep et al. (2016)	PRF + syn. gr.	35 subj/furc	9	7.57 ± 0.50 4.25 ± 0.44	7.65 ± 1.05 3.97 ± 0.16	7.48 ± 0.61 4.51 ± 0.50	-
	RSV + PRF + syn. gr.	35 subj/furc	9	7.51 ± 0.50 3.34 ± 0.48	7.65 ± 1.05 3.02 ± 0.16	7.42 ± 0.60 3.37 ± 0.49	-
	OFD	35 subj/furc	9	7.54 ± 0.56 5.71 ± 0.45	7.34 ± 0.76 5.22 ± 0.91	7.45 ± 0.50 5.82 ± 0.38	-
Lekovic et al. (2003)	PRP + xen. gr. + c. m.	26 subj/furc	9	-	6.86 ± 0.31 2.79 ± 0.32	-	-
	OFD	26 subj/furc	9	-	6.78 ± 0.28 4.297 ± 0.36	-	-
Santana et al. (2009)	comp. gr + PTFE + CPF	30 subj/furc	12	7.06 ± 0.7 4.01 ± 1.0	6.36 ± 1 2.8 ± 1.6	4.85 ± 0.9 2.4 ± 1.3	-
	OFD	30 subj/furc	12	6.65 ± 0.8 6 ± 0.6	5.95 ± 1.3 5.3 ± 1.0	6.10 ± 1.4 5.56 ± 0.8	-
Eto et al. (2007)	syn. gr. + P15+ CPF	12 subj/furc	6–7	12.8 ± 2.7 11.1 ± 2.1 *	3.2 ± 1.5 2.4 ± 0.7	8.5 ± 2.6 6.1 ± 2.3 *	-
	OFD	12 subj/furc	6–7	13.7 ± 2.6 11.6 ± 3.3 *	4.3 ± 2.7 2.7 ± 2.9	8.4 ± 3.2 6.9 ± 1.9	-
Houser et al. (2001)	xen. gr. + c. m.	16 subj/18 furc	6	6.4 4.6	5.9 3.9	-	5.2 3.0
	OFD	11 subj/13 furc	6	6.3 5.7	5.7 5.4	-	5.5 5.7
Chisatzi et al. (2007)	EMD	10 subj/10 furc	6	10.8 9.35	4.75 2.80	4.70 2.80	-
	OFD	10 subj/10 furc	6	10.9 10.00	4.65 3.10	4.60 4.00	-
Serroni et al. (2021)	L-PRF + aut. b. gr.	18 subj/furc	6	6.56 ± 2.455 4.50 ± 2.595	4.61 ± 1.379 2.33 ± 1.029	5.50 ± 1.043 3.22 ± 1.003	-
	aut. b. gr.	18 subj/furc	6	6.83 ± 2.093 4.89 ± 2.324	5.17 ± 0.618 3 ± 0.343	5.11 ± 0.9 3.67 ± 0.97	-
	OFD	18 subj/furc	6	8.33 ± 2.951 7.39 ± 2.570	5.61 ± 1.614 4.39 ± 1.335	6.06 ± 1.731 5 ± 1.283	-
Anderegg et al. (1999)	b. g.	15 subj/furc	6	6.67 ±0.29 3.40 ± 0.21	-	-	-
	OFD	15 subj/furc	6	6.47 ± 0.24 4.07 ± 0.28	-	-	-
Mohamed et al. (2016)	syn. gr. + PRP	11 subj/furc	6	5.7 ± 1.2 4.2 ± 0.9	-	-	-
	OFD	11 subj/furc	6	5.9 ± 1.1 4.6 ± 1.3	-	-	-

VCAL: vertical clinical attachment level, VPD: vertical probing depth, HPD: horizontal probing depth, HCAL: horizontal clinical attachment level, OFD: open flap debridement, CPF: coronally positioned flap, subj: subjects, furc: furcations, al. gr.: allograft, c. m.: collagen membrane, PRF: platelet-rich fibrin, syn. gr.: synthetic graft, RSV: rosuvastatin, PRP: platelet-rich plasma, xen. gr.: xenograft, EMD: enamel matrix derivative, comp. gr.: composite graft, L-PRF: leukocyte and platelet-rich fibrin, aut. b. gr.: autologous bone graft, b. g.: bioactive glass, *: statistically significant, PTFE: polytetrafluoroethylene, P15: inorganic bovine-derived hydroxyapatite matrix/cell-binding peptide.

**Table 7 materials-15-03194-t007:** (A) Changes reported from baseline and follow-up period in studies where non-resorbable membranes were used. (B) Changes reported from baseline and follow-up period in studies where absorbable membranes were used. (C) Changes reported from baseline and follow-up period in studies where blood derivatives were used. (D) Changes reported from baseline and follow-up period in studies where miscellaneous materials were used.

**(A)**							
**Study (Year)**	**Intervention**	**N**	**Follow-up Period** **(Months)**	**Outcome** **(pre-op and post-op)**			
				**VCAL (mm)**	**VPD (mm)**	**HCAL (mm)**	**HPD (mm)**
**Non-Resorbable Membranes**							
Mombelli et al. (1996)	E-PTFE	5 furc	12.5	−0.4 ± 2.07	0.2 ± 1.3	-	0 ± 1 (2 furc open)
	E-PTFE + antibiotic	5 furc	12.5	0.2 ± 1.48	0.8 ± 1.3	-	1.2 ± 1.09
	OFD + antibiotic	5 furc	12.5	0 ± 1.22	1 ± 0.71	-	0.4 ± 1.52
	OFD	5 furc	12.5	−0.8 ± 0.84	0.4 ± 0.55	-	−0.5 ± 0.58 (1 furc open)
Pontoriero et al. (1988) (A)	PTFE	21 subj/furc (buccal)	6	4.1 *_1_	4.5 *	4.1 ± 1.3 *	-
	OFD	21 subj/furc (buccal)	6	1.5 *	2.8 *	1.9 ± 1.3	-
Pontoriero et al. (1988) (B)	PTFE	21 subj/furc (buccal)	6	2.9 *_1_	3.5 *	3.3 ± 1 *	-
	OFD	21 subj/furc (buccal)	6	0.6	2.1 *	2.2 ± 1.1 *	-
Metzler et al. (1991)	E-PTFE	17 subj/furc	6	1 ± 0.9	1.7 ± 0.8	-	-
	OFD	17 subj/furc	6	0.2 ± 0.6	0.9 ± 0.8	-	-
Avera et al. (1998) (A)	PTFE	8 subj/furc (buccal)	9	2 ± 0.63_1_	2.88 ± 0.48_1_	-	-
	OFD	8 subj/furc (buccal)	9	0.5 ± 0.42	1.38 ± 0.65	-	-
Avera et al. (1998) (B)	PTFE	8 subj/furc (lingual)	9	1.5 ± 0.46_1_	2.88 ± 0.55_1_	-	-
	OFD	8 subj/furc (lingual)	9	0.13 ± 0.48	1.25 ± 0.53	-	-
Pontoriero et al. (1995)	E-PTFE + CPF	28 subj/furc (mesial)	6	0.7 *	1.6 *	-	-
	OFD	28 subj/furc (mesial)	6	0.1	1.3 *	-	-
**(B)**							
**Study (year)**	**Intervention**	**N**	**Follow-up period** **(months)**	**Outcome** **(pre-op and post-op)**			
				**VCAL (mm)**	**VPD (mm)**	**HCAL (mm)**	**HPD (mm)**
**Absorbable Membranes**							
Balusubramanya et al. (2012)	v. m. + CPF	11 furc	6	2.18 ± 0.60 *_1_	-	-	1.54 ± 1.04 *
	OFD	11 furc	6	1.09 ± 0.94 *	-	-	1.37 ± 1.12 *
Verma et al. (2011)	aut. periost. gr.	12 subj/furc	6	2.17 ± 0.72 *_1_	-	-	-
	OFD	12 subj/furc	6	0.83 ± 0.72 *	-	-	-
Paul et al. (1992)	c. m.	7 subj/14 furc	6	1.64 ± 0.84	1.50 ± 0.76 *_1_	-	-
	OFD	7 subj/14 furc	6	1.00 ± 1.61	0.86 ± 0.77	-	-
Yukna et al. (1996)	c. m.	27 furc	6–12	0.8 ± 1.4	1.7 ± 1.3 *	-	-
	OFD	27 furc	6–12	0.4 ± 1.8	1.3 ± 1.4 *	-	-
**(C)**							
**Study (year)**	**Intervention**	**N**	**Follow-up period** **(months)**	**Outcome** **(pre-op and post-op)**			
				**VCAL (mm)**	**VPD (mm)**	**HCAL (mm)**	**HPD (mm)**
**Blood Derivatives**							
Bajaj et al. (2013)	PRF	12 (24 furc)	9	2.87 ± 0.85 *_1_	4.29 ± 1.04 *_1_	2.75 ± 0.94 *_1_	-
	PRP	13 (25 furc)	9	2.71 ± 1.04 *_1_	3.92 ± 0.93 *_1_	2.5 ± 0.83 *_1_	-
	OFD	12 (23 furc)	9	1.37 ± 0.58 *	1.58 ± 1.02 *	1.08 ± 0.50 *	-
Kanoriya et al. (2017)	PRF	24 subj/furc	9	3.39 ± 0.49 *_1_,_2_	3.69 ± 0.76 *_1_,_2_	2.86 ± 0.062 *_1_,_2_	-
	PRF + 1% ALN	25 subj/furc	9	4.12 ± 0.6 *_1_	4.4 ± 0.57 *_1_	3.64 ± 0.90 *_1_	-
	OFD	23 subj/furc	9	2.33 ± 0.48 *	2.41 ± 0.77 *	2.04 ± 0.35 *	-
Sharma et al. (2011)	PRF	18 subj/furc	9	2.333 ± 0.485 *_1_	4.056 ± 0.416 *_1_	2.667 ± 0.594 *_1_	-
	OFD	18 subj/furc	9	1.278 ± 0.461 *	2.889 ± 0.676 *	1.889 ± 0.758 *	-
Pradeep et al. (2009)	PRP	20 subj/furc	6	2.50 ± 1.64 *_1_	2.3 ± 1.41 *_1_	2.50 ± 1.17 *_1_	-
	OFD	20 subj/furc	6	0.10 ± 1.10	0.80 ± 1.31	0.80 ± 0.63 *	-
Agarwal et al. (2020)	PRF	15 furc	6	3.55 ± 1.05 *_1_	3.80 ± 0.77 *_1_	-	1.80 ± 0.83 *
	PRF+ al. gr.	15 furc	6	3.90 ± 0.72 *_1_	4 ± 0.79 *_1_	-	1.80 ± 0.41 *
	OFD	15 furc	6	1.35 ± 0.49 *	1.50 ± 0.76 *	-	1.35 ± 0.67 *
Siddiqui et al. (2016)	PRF	15 furc	6	2.40 ± 0.91 *_1_	2.27 ± 1.10 *	2.40 ± 1.06 *_1_	-
	β-TCP	15 furc	6	2.53 ± 0.83 *_1_	2.47 ± 1.51 *	2.27 ± 0.46 *_1_	-
	OFD	15 furc	6	0.93 ± 0.46 *	1.03 ± 0.67 *	0.73 ± 0.46 *	-
**(D)**							
**Study (year)**	**Intervention**	**N**	**Follow-up period** **(months)**	**Outcome** **(pre-op and post-op)**			
				**VCAL (mm)**	**VPD (mm)**	**HCAL (mm)**	**HPD (mm)**
**Miscellaneous Materials**							
Tsao et al. (2006)	al. gr.	9 subj/furc	6	0.1 ± 1	0.9 ± 0.9 *	-	1.2 ± 1.9
	al. gr. + c. m.	9 subj/furc	6	−0.3 ± 1.2	0.7 ± 1	-	1.1 ± 0.5 *
	OFD	9 subj/furc	6	−0.9 ± 1.6	0.1 ± 1.1	-	0.9 ± 1.9
Pradeep et al. (2016)	PRF + syn. gr.	35 subj/furc	9	3.31 ± 0.52 *_1_‚_2_	3.68 ± 1.07 *_1_‚_2_	2.97 ± 0.56 *_1_‚_2_	-
	RSV + PRF + syn. gr.	35 subj/furc	9	4.17 ± 0.70 *_1_	4.62 ± 1.03 *_1_	4.05 ± 0.76 *_1_	-
	OFD	35 subj/furc	9	1.82 ± 0.78 *	2.11 ± 1.25 *	1.62 ± 0.64 *	-
Lekovic et al. (2003)	PRP + xen. gr. + c. m.	26 subj/furc	9	3.29 ± 0.42 *_1_	4.07 ± 0.33 *_1_	-	-
	OFD	26 subj/furc	9	1.68 ± 0.31 *	2.49 ± 0.38 *	-	-
Santana et al. (2009)	comp. gr + PTFE + CPF	30 subj/furc	12	3.05 ± 0.6 *_1_	3.56 ± 0.6 *_1_	3.45 ± 1.3 *_1_	-
	OFD	30 subj/furc	12	0.65 ± 0.6 *	0.6 ± 1 *	0.55 ± 0.7 *	-
Eto et al. (2007)	syn. gr. + P15+ CPF	12 subj/furc	6–7	-	-	-	-
	OFD	12 subj/furc	6–7	-	-	-	-
Houser et al. (2001)	xen. gr. + c. m.	16 subj/18 furc	6	1.8 ± 1.8 *	2 ± 1.7 *_1_	2.2 ± 2.2 *_1_	-
	OFD	11 subj/13 furc	6	0.6 ± 2.06	0.3 ± 1.37	−0.2 ± 1.6	-
Chisatzi et al. (2007)	EMD	10 subj/10 furc	6	1.45 *	1.95 *	1.9 *_1_	-
	OFD	10 subj/10 furc	6	0.9 *	0.9 *	0.6 *_1_	-
Serroni et al. (2021)	L-PRF + aut. b. gr.	18 subj/furc	6	2.139 ± 0.278 *_1_‚_2_	2.515 ± 0.714 *_1_	2.299 ± 0.18 *_1_‚_2_	-
	aut. b. gr.	18 subj/furc	6	1.994 ± 0.276 *_1_	2.150 ± 0.169 *_1_	1.613 ± 0.183 *_1_	-
	OFD	18 subj/furc	6	0.811 ± 0.284 *	1.002 ± 0.714 *	0.866 ± 0.184 *	-
Anderegg et al. (1999)	b. g.	15 subj/furc	6	3.27 ± 0.27 *	-	-	-
	OFD	15 subj/furc	6	2.40 ± 0.24 *	-	-	-
Mohamed et al. (2016)	syn. gr. + PRP	11 subj/furc	6	-	-	-	2.3
	OFD	11 subj/furc	6	-	-	-	1.7

VCAL: vertical clinical attachment level, VPD: vertical probing depth, HPD: horizontal probing depth, HCAL: horizontal clinical attachment level, OFD: open flap debridement, CPF: coronally positioned flap, PRF: platelet-rich fibrin, PRP: platelet-rich plasma, ALN: alendronate, RSV: rosuvastatin, EMD: enamel matrix derivative, PTFE: Polytetrafluoroethylene, E-PTFE: expanded PTFE, subj: subjects, furc: furcations, v. m.: vicryl mesh, periost. gr.: autologous periosteal graft, c. m.: collagen membrane, al. gr.; allograft, β-TCP: beta-tricalcium phosphate, al. gr.: allograft, syn. gr: synthetic graft, xen. gr.: xenograft, comp. gr.: composite graft, L-PRF: leukocyte and platelet-rich fibrin, aut. b. gr.: autologous bone graft, aut. periost. gr.: autologous periosteal graft, b. g.: bioactive glass, A: buccal furcation, B: lingual furcation, _1_: statistically significant difference between test and control group, _2_: statistically sign differences between the test groups, *: statistically significant.

## Data Availability

Publicly available datasets were analyzed in this study.

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
