# Peer review of "Overview of the Effect of Different Regenerative Materials in Class II Furcation Defects in Periodontal Patients"

_materials, 2022, doi:10.3390/ma15093194_

Round 1
Reviewer 1 Report
The follow are some major issues to be addressed in order to improve the quality of the manuscript.
Title
- The title “Overview of the effect of different regenerative materials in Class II furcation defects in periodontal patients.” addresses the content and main message of the paper.
Abstract
- Abstract reflects the content of the paper.
Introduction
- Introduction section is too long and fails to provide a scientific background and explanation of rationale for the review. Perhaps it should be more focused on the clinical studies and meta-analyses that investigated the effect of regenerative materials in affected furcation sites. A relevant citation to these studies should also be included.
- It would be more appropriate to refine information into important definitions and existing knowledge related to the main objective of the study.
- The objective is clear.
Materials & methods
- Please identify the key elements of the research question using framework such as PICO.
- Provide a more detailed description of supplementary searching of relevant studies (e.g. gray literature, hand searching etch).
- Primary outcomes (clinical variables) should be included in this section.
Results
- Results section is too extended and provides a lot of information that should be analyzed at the discussion section.
- The original reasons for study exclusion should be further discussed at this section.
Discussion
- Discussion is weak.
- Discuss limitations of the evidence included in this study (e.g. methodology of included studies).
- Limitations of the review process addressing sources of bias must be included (e.g gray literature, the use of specific databases including the reasons of rejection others like Scopus or Web of Science)
Conclusion is clear.
Author Response
1st Reviewer:
The follow are some major issues to be addressed in order to improve the quality of the manuscript.
Introduction
- Introduction section is too long and fails to provide a scientific background and explanation of rationale for the review. Perhaps it should be more focused on the clinical studies and meta-analyses that investigated the effect of regenerative materials in affected furcation sites. A relevant citation to these studies should also be included.
- It would be more appropriate to refine information into important definitions and existing knowledge related to the main objective of the study.
- The objective is clear.
We thank the reviewer for this valuable comment. We agree that the introduction was on the long side. We have also considered to move some parts to the discussion section. However, we tried to keep the main content of the introduction. Our first aim was to give also an amount of basic information for the reader who is less familiar with these materials and the wide aim of this review. Our decision is corroborated also by the comments of other reviewers who appreciated the information included. At the end we reduced the whole introduction of about 200 words.
We also introduced in the introduction a very valuable previous meta-analysis, as suggested by the reviewer. However, we have preferred to elaborate on this specific and other meta-analyses more extensively in the discussion section instead of the introduction in order not to make the introduction section even longer than what it is now.
We separated the objective and tried to make the rationale behind it even clearer.
Materials & methods
- Please identify the key elements of the research question using framework such as PICO.
We added the PICO format strategy in M&M section as from your request. Thanks for the comment.
- Provide a more detailed description of supplementary searching of relevant studies (e.g. gray literature, hand searching etch).
The sentence “Gray literature was searched in the OpenGrey database. In addition, manual search with the same queries list was performed to retrieve additional articles” was included to clarify the searching process. Thank you.
- Primary outcomes (clinical variables) should be included in this section.
The information you requested are now in M&M section. Thank you.
Results
- Results section is too extended and provides a lot of information that should be analyzed at the discussion section.
- The original reasons for study exclusion should be further discussed at this section.
According to your request, we added the exclusion reasons for the papers after the full-text screening: “From the remaining 63 publications, 5 articles were excluded due to some inconsistencies in data with the agreement of all the authors: Caton et al. [38] was excluded since the data were provided only in charts but not as numerical values. The same reason was considered for Dubrez et al. [40] at least for the primary outcome (VCAL); Lekovic et al. [46] reported the mean ± standard error and not standard deviation; both Wang et al. [60] and Jaiswal et al. [43] reported not reliable data.” Thank you.
Discussion
- Discussion is weak.
We agree with this comment. We have enlarged all the sections and added a part by referring to existing relevant meta-analyses. We also included other suggested recommendations from previous meta-analyses. Next to our recommendation, we also included other recommendations suggested by previous meta-analyses. In addition, we have also added a part of criticism to suggest the reader that mean values are not the only factors leading to the final clinical decision. We have also added a more extensive suggestion for future trials.
- Discuss limitations of the evidence included in this study (e.g. methodology of included studies).
The limitations of the included studies were already described in the Discussion. “Main reasons were the large heterogeneity of the study populations, the differences in the study designs, treatment protocols and primary outcomes, the low to moderate quality of the majority of the studies resulted from the quality assessment (main exceptions were the studies which used blood derivatives) and the scarce availability of multi-armed randomized control trials which compare different materials with each other and with the OFD. In addition, the majority of the studies have a small sample size, a relatively short follow-up (up to 6 to 12 months) and no long-term data on the stability of the clinical results or tooth loss were reported.” Thank you for the comment.
- Limitations of the review process addressing sources of bias must be included (e.g gray literature, the use of specific databases including the reasons of rejection others like Scopus or Web of Science)
According to your request, we asked for a few-days extension for the re-submission in order to make a more complete research also on other databases. We found two more papers we could include which fulfil the inclusion criteria of the present study. We considered also Scopus, Web of Science and the gray literature beside the previous database. Thank you for your suggestion.
Conclusion is clear.

Reviewer 2 Report
The paper – a literature review is a well-organized and well written.
The authors describe clearly and motivate the selection of the subject. The techniques available today for class II Furcation defects treatment are completely described.
Inclusion and exclusions criteria for the articles selected in this Overview are also well selected. The database created after article evaluation, and the selection of the articles are also well defined.
The discussions are not giving us (scientist and practitioners) any advice about which technique is more appropriate for treating Furcation Class II Molar Defects, using the Open Flap Debridement or combining this technique with different regenerative solutions.
The conclusions are reflecting today’s "protocols" in periodontology, where clinician experience and subjective personal choice in regenerative materials selection are the most crucial. Final conclusion is accurate: More studies should be conducted in order to standardized treatments in Periodontology, following different criteria from anatomy – to disease evolution and prognostic.
Author Response
2nd Reviewer:
The paper – a literature review is a well-organized and well written.
The authors describe clearly and motivate the selection of the subject. The techniques available today for class II Furcation defects treatment are completely described.
Inclusion and exclusions criteria for the articles selected in this Overview are also well selected. The database created after article evaluation, and the selection of the articles are also well defined.
The discussions are not giving us (scientist and practitioners) any advice about which technique is more appropriate for treating Furcation Class II Molar Defects, using the Open Flap Debridement or combining this technique with different regenerative solutions.
The conclusions are reflecting today’s "protocols" in periodontology, where clinician experience and subjective personal choice in regenerative materials selection are the most crucial. Final conclusion is accurate: More studies should be conducted in order to standardized treatments in Periodontology, following different criteria from anatomy – to disease evolution and prognostic.
We are grateful for your comment. Although it was described in the Abstract “…the use of regenerative materials may be considered moderately beneficial in the treatment of molars with grade II furcation involvement”, a sentence in the Discussion was also included “The results from the studies included in the present review suggest that surgical therapy combined with the use of regenerative materials can lead to a mild to moderate better clinical outcomes, in particular for VCAL and VPD, in mandibular buccal class II furcations, when compared to OFD alone.”
We also included other suggested recommendations from previous meta-analyses. However, we have also added a part of criticism to suggest the reader that mean values are not the only factors leading to the final clinical decision. We have also added a more extensive suggestion for future trials.

Reviewer 3 Report
This review investigated the outcomes of the use of different regenerative materials managed to treat molars with class II furcation defects in patients with periodontitis in comparison with open flap debridement. The manuscript is well-written. The review is clinically relevant and significant. The reviewer only suggests the abstract should be concised. Therefore, the manuscript can be considered for the publication.
Author Response
3rd Reviewer:
This review investigated the outcomes of the use of different regenerative materials managed to treat molars with class II furcation defects in patients with periodontitis in comparison with open flap debridement. The manuscript is well-written. The review is clinically relevant and significant. The reviewer only suggests the abstract should be concised. Therefore, the manuscript can be considered for the publication.
Thank you for your comment. According to your request, we made the abstract shorter reducing the text under the limit of 300 words.

Round 2
Reviewer 1 Report
-